Association between foot thermal responses and shear forces during turning gait in young adults

Gonzalez Angel E.
Pineda Gutierrez Ana
Kern Andrew M.
Takahashi Kota Z. ktakahashi@unomaha.edu
Department of Biomechanics, University of Nebraska—Omaha , Omaha , NE , United States of America
Abdala Virginia
Electronic publication date: 2021 Jan 18
Publication date: 2021
Volume: 9
Electronic Location ID: e10515
Received 2020 Jun 17; Accepted 2020 Nov 17
Copyright: ©2021 Gonzalez et al.
Copyright year: 2021
Copyright holder: Gonzalez et al.
License: This is an open access article distributed under the terms of the Creative Commons Attribution License, which permits unrestricted use, distribution, reproduction and adaptation in any medium and for any purpose provided that it is properly attributed. For attribution, the original author(s), title, publication source (PeerJ) and either DOI or URL of the article must be cited.
License URL: https://creativecommons.org/licenses/by/4.0/

Keywords: Feet, Locomotion, Biomechanics, Curved path walking, Thermoregulation

Funding: the National Institutes of Health P20GM109090 This work was supported by the National Institutes of Health [P20GM109090 to Kota Takahashi]. The funders had no role in study design, data collection and analysis, decision to publish, or preparation of the manuscript.

==============================
Background

The human foot typically changes temperature between pre and post-locomotion activities. However, the mechanisms responsible for temperature changes within the foot are currently unclear. Prior studies indicate that shear forces may increase foot temperature during locomotion. Here, we examined the shear-temperature relationship using turning gait with varying radii to manipulate magnitudes of shear onto the foot.

Methods

Healthy adult participants (N = 18) walked barefoot on their toes for 5 minutes at a speed of 1.0 m s−1 at three different radii (1.0, 1.5, and 2.0 m). Toe-walking was utilized so that a standard force plate could measure shear localized to the forefoot. A thermal imaging camera was used to quantify the temperature changes from pre to post toe-walking (ΔT), including the entire foot and forefoot regions on the external limb (limb farther from the center of the curved path) and internal limb.

Results

We found that shear impulse was positively associated with ΔT within the entire foot (P < 0.001) and forefoot (P < 0.001): specifically, for every unit increase in shear, the temperature of the entire foot and forefoot increased by 0.11 and 0.17 °C, respectively. While ΔT, on average, decreased following the toe-walking trials (i.e., became colder), a significant change in ΔT was observed between radii conditions and between external versus internal limbs. In particular, ΔT was greater (i.e., less negative) when walking at smaller radii (P < 0.01) and was greater on the external limb (P < 0.01) in both the entire foot and forefoot regions, which were likely explained by greater shear forces with smaller radii (P < 0.0001) and on the external limb (P < 0.0001). Altogether, our results support the relationship between shear and foot temperature responses. These findings may motivate studying turning gait in the future to quantify the relationship between shear and foot temperature in individuals who are susceptible to abnormal thermoregulation.

Introduction

Thermoregulation is a vital function of the nervous system in response to cold or heat stress which allows internal core temperature to converge to a stable temperature to maintain homeostasis. Proximal areas of the body are usually characterized by non-glabrous skin (hairy skin) and lack of arteriovenous anastomoses—features that allow proximal areas to behave as an insulator (Romanovsky, 2014). Heat exchange organs characterized by dense vascularization, presence of arterio-venous anastomoses and a large surface-to-volume ratio are usually found on distal segments of the body (e.g., foot). These features allow the foot to act as a radiator to explore the environment and exchange heat with it through interaction (via conduction) and dissipation of heat (Romanovsky, 2014). Furthermore, other mechanisms like sweating and vasoconstriction or vasodilation can assist in thermoregulation of the foot (Cheung, 2015; Taylor et al., 2009). The ability to regulate foot temperature may be especially important during or after locomotion activities since studies on healthy adults have demonstrated temperature differences from pre- to post-activity (Carbonell et al., 2019; Najafi et al., 2012; Reddy et al., 2017; Shimizaki & Murato, 2015). Though these temperature fluctuations may not be a problem for healthy individuals, potential negative consequences may arise for individuals that are unable to maintain within a desired thermal state within the foot. For example, high tissue temperature is a major factor for diabetic foot ulcer occurrence (Armstrong et al., 2007). Thus, physically-demanding locomotor tasks may predispose the foot tissue to harmful consequences such as ulceration within neuropathic individuals.

The thermal response of the foot has been shown to significantly change in response to different locomotor tasks or footwear in healthy adult participants, including both increase and decrease in temperature from pre- to post-activity. For instance, foot temperature increases by ∼8.2 °C in the heelpad after 30 min of running (Shimazaki & Murato, 2015) and ∼4.6 °C within the entire foot after 45 min of walking (Reddy et al., 2017). Likewise, neuropathic individuals also demonstrate an overall increase in foot temperature as speed increases during shod-walking within the affected and non-affected limbs (Najafi et al., 2012). Foot temperature is also known to be affected by wearing socks and shoes (Luximon, Ganesan & Younus, 2017). Similarly, the type of sock material has been mentioned as another possible factor affecting heat dissipation within the foot (Van Roekel, Poss & Senchina, 2014). Additionally, walking speed can increase the rate of foot temperature increase. However, a relative plateau occurs for various levels of walking cadence where the final temperature does not change (Reddy et al., 2017). There are also studies that have found a decrease in foot temperature from pre- to post-activity, specifically in activities of relatively short duration. For example, in patients with Charcot neuropathy, it has been reported that after 50 steps of walking, the plantar temperature decreases compared to the initial baseline temperatures (although the temperature decrease is less as more steps are taken) (Najafi et al., 2012). Another study found that walking 100 m can decrease foot temperature by ∼1.3 °C in healthy adults and by ∼2.0 °C in patients with diabetes (Carbonell et al., 2019). Although these different factors (e.g., form of locomotion, speed, duration, and insulation) are known to affect the temperature response (either increase or decrease), the primary mechanism responsible for this change in foot temperature during locomotion is less clear.

The foot-to-ground mechanical interaction is a potential predictor of the temperature change within the foot during walking. The foot has been described as a dynamic system that can perform multiple functions: accommodate to the variations in the external environment, act as a spring and lever for push-off during gait (Farris, Birch & Kelly, 2020; Ray & Takahashi, 2020), and to enable body weight to be carried with adequate stability (Kelly et al., 2016; Kelly, Lichtwark & Cresswell, 2018; Kirby, 2017; Rodgers, 1995). A combination of these functions enables the foot to behave as a shock absorber, potentially allowing mechanical work to dissipate as thermal energy. In fact, more recent approaches using foot modeling have revealed that the foot structures can deform and dissipate mechanical energy during walking (Honert & Zelik, 2019; Papachatzis et al., 2020; Takahashi, Worster & Bruening, 2017) and running (Kelly, Lichtwark & Cresswell, 2018). Given that the foot dissipates mechanical energy, the increases in foot temperature may be a result of the repetitive forces encountered during walking.

While mechanical stress that tends to compress the foot vertically was previously implicated as a causative factor in the change of foot temperature (Houghton, Bower & Chant, 2013), more recent studies utilizing custom-built pressure-shear plates reveal shear has a stronger predictive value (Yavuz, 2013; Yavuz et al., 2014; Yavuz et al., 2017). Custom-built pressure-shear plates have enabled pressure and shear forces to be measured concurrently during walking within the foot. Local shear stress within neuropathic individuals has been observed, in which the lateral regions of the metatarsals experience approximately two-fold higher shear stress compared to the medial and toe regions of the forefoot (Perry, Hall & Davis, 2002). Within healthy individuals, these areas of peak shear stress are related to areas of higher temperature elevations (Yavuz et al., 2014), providing empirical support for the link between shear and temperature responses. However, technical challenges prevent the measurement of shear stress without the use of custom-built pressure-shear plates (Perry, Hall & Davis, 2002; Yavuz et al., 2017). A major limitation of standard force plates is that only a single net force can be measured, which neglects the presence of opposing shear forces that act within the foot (Bruening et al., 2010). Although the difficulties of directly measuring plantar shear stress preclude its use in large thermoregulation studies, certain types of locomotion could be used to minimize the opposing shear forces underneath the foot. Specifically, toe-walking may be one viable method of isolating the shear forces under the forefoot and reducing the presence opposing shear forces (e.g., between the rearfoot and forefoot) when using a standard force platform (Bruening et al., 2010).

A potential way to investigate the causal relationship between shear forces and temperature may be through turning gait. An increase of mediolateral ground reaction forces has been revealed as one fundamental aspect during different types of turning (Orendurff et al., 2006). In support of this, the mediolateral impulse is reported to have the highest pronounced differences during turning compared to straight-line walking (Orendurff et al., 2006). Furthermore, the external limb (limb farther from the center of the curved path) is reported to experience greater shear forces compared to the internal limb (limb closer to the center of the curved path), due to a greater force needed to orient the body towards the turning direction (Orendurff et al., 2006). Turning at a sharper angle (i.e., smaller radius) may in turn result in higher mediolateral forces to orient the body in a new path.

The purpose of this study was to investigate the thermal response of the foot to varying magnitudes of shear forces during barefoot curved-path walking. The study utilized toe-walking during continuous turning with varying radii as a means of influencing the differing magnitude of shear forces encountered at the forefoot region (metatarsal-phalangeal joints and toes). Due to the mechanical requirement of the task, greater shear forces were expected when turning with a smaller radius, and greater shear was expected on the external foot compared to the internal foot (Orendurff et al., 2006). It was hypothesized that the foot temperature increase would be related to a greater shear force. It was also hypothesized that the external foot, due to greater shear forces, would experience higher temperature increase compared to the internal foot.

Methods

Participants

Nineteen healthy young adults (N = 9 females & 10 males, age: 25.95 ± 4.25, height: 170.66 ± 11.67 cm, mass: 73.03 ± 14.86 kg) participated in this study, conducted at the University of Nebraska at Omaha under the approval of the Internal Review Board of the University of Nebraska Medical Center (IRB Protocol #: 146-19-EP). Written consent from each participant was taken prior to the experiment. To be considered healthy, all participants were within a healthy Body Mass Index (BMI) range: 18.5 to 29.9 kg m−2 (high-end BMI allowing for individuals with low body fat %). Participants ranged in activities from strength training, running, walking, biking, soccer, basketball, baseball, swimming, racketball, climbing, fencing, hiking, to yoga. Different intensities and days were reported for each participant. Participant background information, as well as anthropometrics, were obtained prior to commencing the experiment. A power analysis was conducted based on a study by Patterson et al., analyzing the effect of additional load carriage on foot temperature (Patterson et al., 2018). The mean temperature effect of the 0% additional body weight condition (2.2 °C) and 30% additional body weight condition (3.3 °C) on the plantar aspect of the foot was used to determine appropriate sample size. A sample size of 24 participants provided 80% power to detect similar differences to the Patterson study (effect size = 0.61), with the significance level set at α = 0.05.

Experimental protocol

Each participant attended a single ∼3 h session in which all experimental procedures were implemented. Participants were asked to wear lower extremity loose clothing for proper marker placement on each foot. Prior to marker placement, participants were asked to remove their socks and shoes. 14 retro-reflective markers were placed on each participants’ feet, adapted from a previous study (Bruening, Cooney & Buczek, 2012). Participants had a 20-minute acclimatization period to adjust their feet to room temperature before each walking condition. Participants were then asked to walk barefoot on over-ground trials on curved paths of varying radii. The curved paths were fixed in line with a standard force plate (AMTI OPT400600-1000m AMTI Inc., Watertown, MA) placed on the floor, recorded by 17 infrared cameras (Motion Analysis Corp., Rohnert Park, CA).

The radii for the curved paths were: 1.0 m, 1.5 m, and 2.0 m (Fig. 1). Walking speed was controlled by using a speed light timing system with a speed of 1.0 m s−1 and a permissible deviation of ±0.05 m s−1 (Dashr Elite Kit, Lincoln, NE). One timing gate, consisting of a laser module and a reflector, determined the walking speed after each lap around the curved path via Bluetooth on an application (Dashr App) to a connected mobile device. The timing gate was placed in a perpendicular direction on the internal and external sides of the curved path. Participants were immediately instructed after each lap completed to speed up or slow down if they fell outside of the permissible speed deviation.

Figure 1 The experimental setup involved pre- and post-temperature measurements (A) as participants (n = 18) walked on their toes along a curved path with varying radii (B).

The order of the radii conditions (1.0, 1.5, and 2.0 m) was randomized for each participant. Participants were instructed to walk on their toes around each curved path while maintaining a fixed speed (1.0 m s−1). A force platform embedded on the floor was used to quantify shear forces acting on the foot.

Participants walked barefoot for five minutes on each of the curved paths in a randomized order. Participants were instructed to walk on their toes for the entirety of each trial, monitored by an additional investigator. Following each walking trial, post-temperature of the foot was assessed on the plantar aspect of the foot (Fig. 1). Plantar temperature measurements were gathered by utilizing a thermal imaging camera with a resolution of 464 × 348 pixels capable of detecting temperatures ranging from −20 °C to 1500 °C with accuracy of ±2.0% from the temperature reading (FLIR T540sc, Wilsonville, OR). Prior to taking thermal images, the camera was given a 20 min period to allow the sensors to calibrate to the surrounding temperature (23.42 ± 1.22 °C) and humidity (42.42 ± 10.64%). The images were acquired with the camera approximately 1.0 m apart from the plantar aspect of the foot at an emissivity setting of 0.98. For transparency in the data acquisition and data reporting of thermal imaging data, we uploaded a supplementary checklist (Fig. S1) of recommended guidelines set forth by a prior study (Moreira et al., 2017). We note that we did not address a few of the guidelines, including specific instructions regarding participants’ diet, alcohol consumption, caffeine intake prior to data collection, or other extrinsic factors such as physical activity prior to data collection. While this limitation can hinder between-participant comparisons, most of our primary analyses are within-participant comparisons.

Following the temperature measurement, the foot was sanitized by applying 70% isopropyl alcohol to remove any dust accumulated from the lab floor. Participants were then given 20 min of rest to allow the foot to recover to its baseline temperature. Prior studies have used resting times ranging from 5 min (Najafi et al., 2012; Rahemi et al., 2017; Van Netten et al., 2013), 10 min (Jimenez-Perez et al., 2020; Quesada et al., 2015; Reddy et al., 2017; Schmidt, Germano & Milani, 2017; Yavuz et al., 2014), and 15 min (Armstrong & Lavery, 1997). Floor temperature measurements were taken to account for a potential confounding factor by placing contact probes (Extech SDL200, Extech Instruments, Waltham, MA) onto the floor surface until relatively stable temperatures were reached (15–20 s).

Data analysis

Temperature change (ΔT) was analyzed as the difference between post- and pre-walking temperature values from each condition. The entire foot region was defined by manually tracing the outline of each foot (Fig. 2), performed by one researcher using ResearchIR (FLIR, Wilsonville, OR). The forefoot region was defined by manually tracing and encapsulating: the toes and the first and fifth metatarsal heads (Fig. 2). The lateral side of the fifth metatarsal head and the medial side of the first metatarsal head was used to visually define the proximal end of the forefoot region. In order to analyze temperature values from each region, custom MATLAB code (Mathworks, Natick, MA, USA) was used.

Figure 2 A representative image of one participant’s pre- (A) and post- (B) temperature measurements.

The outlines (manually traced for visualization) represent the areas to gather the entire foot (white) and forefoot (gray) temperature. To create a contrast difference between the foot and background temperature, a cold towel was placed above the subject’s leg (no contact between towel and limb) before taking the thermal image.

Each participant’s kinetic and kinematic data were processed and exported using Cortex software (Motion Analysis Corporation, Rohnert Park, CA, USA). Kinetic data were sampled at 1000 Hz and kinematic data were sampled at 100 Hz. Kinetic data were filtered at 25 Hz and kinematic data were filtered at 6 Hz with a 2nd order low-pass Butterworth filter. To express horizontal ground reaction forces in a local coordinate system, a multi-segment foot model was used, adapted from a previous study (Bruening, Cooney & Buczek, 2012). The horizontal ground reaction forces (along the anteroposterior (AP) and mediolateral (ML) axes) were obtained from the force plate and transformed from the laboratory-based coordinate system to a local coordinate system affixed to the foot. The AP axis of the local coordinate system was defined by the component of the longitudinal axis of the foot that was parallel with the ground, where the longitudinal foot axis was defined by the vector from the midpoint between the medial and lateral malleoli markers to the midpoint between the first and fifth metatarsal head markers. The ML axis of the local coordinate system was orthogonal to the AP axis, parallel with the ground.

To determine the contribution of contact time with the floor relative to stride time (duty factor), a full gait cycle was needed and thus a previously established method for determining heel strike and toe-off based on target pattern recognition was used (Stanhope et al., 1990). The impulse from the ML and AP vectors of the ground reaction force (normalized to body weight) as well as the resultant of these two vectors were analyzed for both limbs using Visual 3D software (C-Motion, Germantown, MD, USA) by utilizing the local coordinate system affixed to each foot. The impulse from the resultant shear forces was calculated by integrating the entire time series over the stance phase. Shear impulse was extrapolated to the entirety of the walking trial (5 min). In order to extrapolate the shear data, duty factor (i.e., ratio of floor contact time to stride) from each participant was multiplied by the total time to estimate the total time spent in the stance phase for each radius condition. The product of shear impulse per step and estimated floor contact time was used to gather accumulated shear impulse per five minutes of walking.

One participant was omitted from the study as technical difficulties prevented the collection of their kinetic and kinematic data. Due to this difficulty, all statistical analyses were performed on 18 participants (nine females and nine males).

Secondary analyses

We analyzed additional variables that could affect ΔT, including work done by the foot, free moment, and baseline temperatures. A unified deformable power analysis was utilized to calculate the power contributions of all structures distal to the hindfoot (i.e., entire foot) (Takahashi, Kepple & Stanhope, 2012; Takahashi, Worster & Bruening, 2017). Mechanical work done by the foot was determined by integrating the distal-to-hindfoot power with respect to time during stance. The free moment angular impulse was calculated by integrating the free moment of the force plate during the stance. Free moment was normalized by body weight × height (Creaby & Dixon, 2008; Holden & Cavanagh, 1991; Milner, Davis & Hamill, 2006), and distal-to-hindfoot power was normalized by body mass. Foot mechanical work and free moment angular impulse were extrapolated to the entirety of the walking trial (5 min).

After completing all of the walking trials, the participants performed additional walking trials to quantify foot contact area, which is required to estimate shear stress (defined as the peak resultant shear force over the contact area with the ground). The contact area was gathered by capturing a thermal foot imprint using the thermal imaging camera. The foot imprints were placed on top of a leather cloth-back material by having the participants walk on their toes an additional lap on the curved path for each foot and each radius. To create a clear contrast difference between the foot thermal imprint and leather material, participants had their feet passively warmed using warm water (∼40 °C). After the participant had their feet dried, they were asked to walk barefoot on their toes over the curved path containing the leather material for one lap for each radius. A custom MATLAB code was utilized to gather the average contact area. Briefly, the code reads the temperature data from individual pixels within the FLIR thermal image and sets a threshold gradient between the leather material and foot thermal imprint to detect the area of the forefoot in contact with the leather material. During post-processing, we discovered that some participants’ feet did not create a clear temperature gradient, most likely since the ground contact time was not long enough for the leather material to accumulate heat from the foot. With this technical difficulty, only ten participants were analyzed for the stress data.

Statistical analysis

To analyze differences in shear impulse and ΔT of the entire foot and forefoot across radii conditions and limbs, two-way repeated-measures ANOVA was performed in SPSS (IBM, Armonk, NY). When a significant main effect was found, Fishers’ least significant difference method was used for pair-wise comparisons. Data were reported as means and standard deviations. Statistical significance was defined as α < 0.05.

A linear mixed effects (LME) model analysis was performed in R (R Core Team, 2019) using the extrapolated data to determine the shear-thermal relationship. Although more simple statistical tests (e.g., repeated measures ANOVA) are usually enough, more complex structures require more complex models (Zuur et al., 2009). The LME model allows for multiple fixed effects in addition to controlling for non-independence among data points (by utilizing participants as a random effect). The statistical model was able to determine whether potential confounding variables could potentially influence foot temperature.

A hierarchical approach was used to test for significance, where all potential confounding factors and factors of interest were initially included within the model. Terms were removed from the final model when non-significance was yielded. A likelihood ratio test was performed on two models to determine the effect of each variable. That is, the temperature data were analyzed by comparing a full model as a function of the fixed effects as well as the random effects to a reduced model (subtraction of fixed effects) to determine whether the fixed effect improved upon the model. Additionally, the coefficient of determination was determined to quantify the goodness-of-fit of the fixed effects. The variance explained (R2) reported here were calculated according to Nakagawa & Schielzeth (2012). The variance explained for the fixed effects was reported as marginal R2 values and the variance explained for the full model (i.e., fixed effects plus the random effects) are reported as conditional R2 values.

Results

Testing for assumptions

Shapiro Wilk’s test was performed to check for normality of residuals (entire foot model: P = 0.40, forefoot model: P = 0.60). Homoscedasticity and linearity were confirmed visually via a residual plot. Independence of the data was accounted for by having participants input as a random effect into the statistical model.

Effect of radius condition/limb on shear forces

The resultant shear impulse (extrapolated to 5 min of walking) increased as the radii condition decreased (df = 2, F = 14.77, P < 0.0001) with significant differences between the smallest (1.0 m) radii (15.34 ± 2.46 N Bw−1 s) and medium (1.5 m) radii (14.39 ± 2.04 N Bw−1 s) (P < 0.01), smallest radii and largest (2.0m) radii (13.94 ± 2.21 N Bw−1 s) (P < 0.001), and medium radii and largest radii (P = 0.04) (Fig. 3). The ML impulse (extrapolated to 5 min of walking) increased as the radii condition decreased (df = 2, F = 66.74, P < 0.0001) with significant differences between the smallest radii (10.30 ± 1.74 N Bw−1 s) and medium radii (8.31 ± 1.32 N Bw−1 s) (P < 0.0001), smallest radii and largest radii (7.17 ± 1.19 N Bw−1 s) (P < 0.0001), and medium radii and largest radii (P < 0.0001). The AP impulse (extrapolated to 5 min of walking) decreased as the radii condition decreased (df = 2, F = 12.80, P < 0.0001) with significant differences between the smallest radii (9.15 ± 2.33 N Bw−1 s) and medium radii (10.44 ± 1.57 N Bw−1 s) (P < 0.01), smallest radii and largest radii (10.82 ± 1.95 N Bw−1 s) (P < 0.001), and medium radii and largest radii (P = 0.04).

Figure 3 The anteroposterior (AP) (A) and mediolateral (ML) (B) components of the ground reaction forces (time-normalized to stance phase) were utilized to gather resultant shear impulse extrapolated to the 5 min walking trial for each limb within each radii condition

AP and ML forces are relative to the coordinate system of each foot, where medial forces on the external limb and lateral forces on the internal limb indicate forces directed towards the center of the radii, respectively. The resultant shear impulse was greater within the external limb compared to the internal limb (p < 0.0001, denoted by an asterisk) and greater as the radii condition decreased (p < 0.0001, denoted by a + symbol) (Fig. 2C). Pairwise comparisons are denoted by horizontal lines above each condition. (Values are means ± S.D.).

The external foot (15.96 ± 2.52 N Bw−1 s) experienced significantly greater accumulated shear impulse compared to the internal foot (13.16 ± 2.42 N Bw−1 s) (df = 1, F = 48.18, P < 0.0001). The ML impulse was greater within the external limb (11.17 ± 2.26 N Bw−1 s) compared to the internal limb (6.01 ± 2.43 N Bw−1 s) (df = 1, F = 76.96, P < 0.0001). There were no significant differences in AP impulse between limbs (df = 2, F = 3.43, P = 0.08).

Effect of radius/limb on temperature pre (baseline) and post walking

Order of the trials had no significant effect on baseline temperature within the entire foot (df = 2, F = 0.76, P = 0.48) or forefoot (df = 2, F = 0.54, P = 0.59). However, a lower baseline temperature was observed within the entire foot as the radius got smaller (df = 2, F = 5.36, P < 0.01) with significant differences between the smallest radius (25.88 ± 2.12 °C) and medium radius (26.37 ± 2.26 °C) (P = 0.04) and between the smallest and largest (26.55 ± 2.30 °C) radius (P < 0.01) and no differences between the medium and largest radii (P = 0.45) (Fig. S2). Baseline temperature differences were not observed within the forefoot (df = 2, F = 3.09, P = 0.06). Baseline temperature differences were not observed between external and internal limbs within the entire foot (df = 1, F = 0.02, P = 0.90) nor forefoot (df = 1, F = 0.58, P = 0.46). No significant differences in post-temperature were determined between radius within the entire foot (df = 2, F = 0.50, P = 0.61) or within the forefoot (df = 2, F = 0.04, P = 0.96). No differences in post-temperature were determined between limbs within the entire foot (df = 1, F = 3.96, P = 0.06) nor forefoot (df = 1, F = 2.97, P = 0.10).

Effect of radius condition/limb on foot temperature change from pre and post walking (ΔT)

Across all trials, ΔT on average was a negative value, indicating that foot temperature decreased after the toe-walking trials. As the radius decreased, ΔT increased (i.e., less negative) within the entire foot (df = 2, F = 8.36, P < 0.01) with significant differences between the smallest radius (−0.45 ± 1.88 °C) and medium radii (−0.87 ± 1.21 °C) (P = 0.01), smallest radius and largest radius (−1.0 ± 1.06 °C) (P < 0.014) (Fig. 4). No difference in entire foot ΔT between the medium radii and largest radii were determined (P = 0.35). As the radius decreased, ΔT increased within the forefoot (df = 2, F = 5.58, P < 0.01) with significant differences between the smallest radius (−0.25 ± 1.81 °C) and medium radii (−0.86 ± 1.85 °C) (P = 0.04), smallest radius and largest radius (−1.09 ± 1.79 °C) (P < 0.01). No difference in forefoot ΔT between the medium radii and largest radii were determined (P = 0.39). There was a greater ΔT within the external limb (−0.63 ± 1.06 °C) compared to the internal limb (−0.92 ± 1.10 °C) in the entire foot (df = 1, F = 17.16, P < 0.01). There was a higher ΔT within the external limb (−0.53 ± 1.61 °C) compared to the internal limb (−0.93 ± 1.78 °C) within the forefoot region (df = 1, F = 11.47, P < 0.01).

Figure 4 Difference between post- and pre-walking temperatures (ΔT) were analyzed for subjects’ (n = 18) between limbs and radii conditions for the entire foot (A) and forefoot (B) region within all subjects.

ΔT was greater (i.e., less negative) as the radii decreased within the entire foot (p < 0.01) and within the forefoot (p < 0.01) (significant radii effect denoted by an asterisk). Greater ΔT was observed in the external limb compared to the internal limb within the entire foot (p < 0.01) and in the forefoot (p < 0.01) (significant limb effect denoted by a + symbol). Pairwise comparisons are denoted by horizontal lines above each condition. (Values are means ± S.D.).

Shear-thermal relationship

Factors significantly affecting foot temperature were included within the final linear mixed effects model. Significant fixed effects affecting the final model included extrapolated shear impulse and gender. Participants were utilized as a random effect within the model. Shear impulse affected entire foot ΔT (χ2(1) = 20.23, P < 0.0001), increasing temperature by about 0.11 ± 0.10 °C for every unit of shear impulse normalized to body weight (Eq. (1)) (Fig. 5). Shear impulse also affected forefoot ΔT (χ2(1) = 17.23, P < 0.0001), increasing temperature by about 0.17 ±0.16 °C for every unit increase of shear (Eq. (2)). (1) EntirefootΔT=0.11x−1.82

(2) ForefootΔT=0.17x−2.51.

Figure 5 A linear mixed models analysis was used to determine the relationship between temperature difference between post- and pre- walking (ΔT) and shear within the entire foot (A) and forefoot (B) of participants (n = 18).

Resultant shear impulse (extrapolated to the 5 min walking trial) was a significant predictor of entire foot ΔT (p < 0.0001) and forefoot ΔT (p < 0.0001). A greater slope for forefoot ΔT was predicted compared to entire foot ΔT. (Open circle = external limb, X = internal limb; blue = 1.0 m, red = 1.5 m, and green = 2.0 m). Marginal variance (i.e., R2 for our fixed effects) was calculated for both regions of the foot.

For the final model (temperature as a function of shear impulse and gender), the coefficient of determination for the fixed effects were: marginal R2 = 0.32 for the entire foot and marginal R2 = 0.24 for the forefoot region. The coefficients of determination for the random effects were: conditional R2 = 0.83 for the entire foot and conditional R2 = 0.81 for the forefoot region. When analyzing the two shear components separately during turning, AP impulse was a significant predictor of entire foot ΔT (P = 0.03) but not forefoot ΔT (P = 0.10). The ML impulse was a significant predictor of entire foot ΔT (P < 0.0001) and forefoot ΔT (P < 0.0001). The limb side was not a significant predictor of entire foot ΔT (P = 0.85) nor forefoot ΔT (P = 0.46).

Effect of baseline temperature on changes in foot temperature (ΔT)

An additional mixed model was used to examine the effect of baseline temperatures on ΔT, where significant fixed effects included extrapolated shear impulse, gender, and baseline temperature. Participants were utilized as a random effect within the model. Even with the inclusion of baseline temperatures, shear impulse continued to be a significant predictor of ΔT. Within the entire foot, shear impulse significantly (χ2(1)= 19.48, P < 0.0001) increased temperature by about 0.07 ± 0.06 °C for every unit of shear impulse normalized to body weight (Eq. (3)). Shear impulse also affected forefoot ΔT (χ2(1) = 23.03, P < 0.0001), increasing temperature by about 0.12 ± 0.10 °C for every unit of shear (Eq. (4)). (3) EntirefootΔT=0.07x−9.79.

(4) ForefootΔT=0.12x−19.62

When accounting for baseline temperature, the coefficient of determination for the fixed effects were: marginal R2 = 0.84 for the entire foot and marginal R2 = 0.81 for the forefoot region. When accounting for baseline temperature, the coefficients of determination for the random effects were: conditional R2 = 0.94 for the entire foot and conditional R2 = 0.96 for the forefoot region.

Additional factors affecting changes in foot temperature (ΔT)

To analyze the effect of potential confounding factors on foot temperature change, a hierarchical approach was utilized (i.e., factors were removed when non-significance was detected). The fixed factors analyzed were: net work, free moment, lab surface temperature, limb side, gender, and extrapolated shear impulse. Participants were utilized as a random effect within the model. Additionally, the effect of shear stress on foot temperature was analyzed separately utilizing significant factors (i.e., including gender as a fixed factor as well).

Net work distal to the hindfoot was not a significant predictor of ΔT within the entire foot (P = 0.87) nor in the forefoot (P = 0.36) (Fig. S3). Free moment angular impulse was also not a significant predictor of entire foot ΔT (P = 0.47) nor forefoot ΔT (p = 0.56) (Fig. S4). Shear stress measured within the forefoot (N = 10) was a significant predictor of entire foot ΔT (P < 0.01) and forefoot ΔT (P = 0.01).

Gender was a significant predictor of ΔT where males had a lower estimated entire foot ΔT relative to females (χ2(1) = 6.36, P = 0.01). Within the entire foot, the pre and post temperature values for females were 25.67 ± 1.14 °C and 25.47 ± 0.96 °C, respectively, and the values for males were 26.86 ± 2.78 °C and 25.52 ± 1.62 °C, respectively. Likewise, males had a lower estimated forefoot ΔT relative to females (χ2(1) = 4.01, P = 0.04). Within the forefoot, the pre and post temperature values for females were 24.60 ± 1.44 °C and 24.61 ± 0.83 °C, respectively, and the values for males were 26.58 ± 3.66 °C and 25.10 ± 1.89 °C, respectively. Lab surface temperature was not a significant predictor of entire foot ΔT (χ2(1) = 4.24, P = 0.11) nor forefoot ΔT (χ2(1) = 1.28, P = 0.44), indicating that the lab surface did not affect ΔT differently across participants or radii conditions.

Discussion

The purpose of this study was to investigate the thermal response of the foot to varying magnitudes of shear forces during continuous turning while toe-walking. Shear forces were greater on the external foot (compared to internal foot), and were greater when turning with a smaller radius (Fig. 3). In support of our hypothesis, we found that ΔT was associated with greater magnitudes of shear impulse (Fig. 5). Furthermore, there was greater shear-temperature sensitivity for the forefoot region than the entire foot, providing further support that shear forces affect temperature as shear was constrained to the forefoot region during toe-walking. Moreover, our hypothesis that the external foot would experience greater temperature increase (compared to internal foot) was supported, suggesting that the foot temperature response is sensitive to between limb changes in shear. However, we note that the foot temperature did not necessarily increase from pre to post-activity. Rather, the foot temperature, on average, decreased following walking, and the smaller radii conditions led to a smaller decrease in foot temperature (i.e., greater ΔT with a smaller radius). We considered other mechanical factors besides shear, such as free moment and net work done by the foot; however, these factors did not significantly affect foot temperature responses. Overall, our results confirm prior studies using shear-sensing platforms (Yavuz, 2013; Yavuz et al., 2014), in which we observed significant associations with shear forces and foot temperature responses using a within-participant design protocol involving turning gait. Furthermore, this study supports the use of a readily available standard force plate to quantify the shear-thermal relationship.

Prior studies have found either increase in ΔT (i.e., feet getting warmer) (Luximon, Ganesan & Younus, 2017; Reddy et al., 2017; Shimazaki & Murato, 2015) or a decrease in ΔT (i.e., colder) (Najafi et al., 2012; Caronelle et al., 2018) between pre- and post-activity. Our study, on average, found the ΔT values as negative values. While the explanatory factor behind the contrasting results is unclear, one possibility is that the lab floor temperature (22.9 ± 0.89 °C) could have decreased the temperature overall, since the entire foot (26.26 ± 2.20 °C) and forefoot (25.59 ± 2.96 °C) of participants were generally warmer at baseline. A prior study has shown that floor temperature affects the skin temperature of the foot during quiet standing (Song, 2008). While our statistical analyses revealed that the lab floor did not affect ΔT of the entire foot nor forefoot among participants or across radii conditions, this result was likely due to the small variability in lab floor temperature. In other words, the lab floor affected ΔT similarly across participants and radii conditions. Thus, the lab floor temperature would not affect our main findings regarding the association between shear forces and ΔT.

Baseline temperatures may have additionally influenced our main finding regarding the association between shear force and ΔT. Greater baseline temperature values were observed before walking on the largest radius compared to the smallest radius. A prior study has found an inverse relationship between baseline temperature and ΔT after walking (Reddy et al., 2017), thus our findings related to increases in ΔT with smaller radius (i.e., greater shear) could be partially explained by the differences in baseline temperatures. These unexpected differences in baseline temperature could not be explained by the order of the trials, since there was no main effect of trial order on baseline temperature. While we cannot fully explain the differences in baseline temperature, we performed a secondary analysis to examine the combined effect of baseline temperature and shear forces on ΔT using a linear mixed model. This secondary analysis revealed that both baseline temperature (P < 0.0001) and shear forces (P < 0.0001) are significant predictors of ΔT. In other words, while baseline temperature can affect differences in ΔT across the various radius conditions, our main conclusions regarding the relationship between shear forces and ΔT are still supported.

While the primary hypothesis of this study was that foot temperature responses were related to shear forces, we also explored alternative mechanisms such as the foot’s mechanical work and free moment. The structures surrounding the foot have been found to dissipate mechanical energy (i.e., produce negative net work) during the stance phase of walking (Takahashi, Worster & Bruening, 2017; Papachatzis et al., 2020). This net dissipative behavior of the foot may be manifested as thermal energy and lead to an increase in foot temperature. Through our secondary analyses, we found that the foot’s net negative work was not a significant predictor of temperature increase. Additionally, we found that the free moment was also not a significant predictor of ΔT. Our protocol, designed to determine the relationship between shear and foot ΔT, did not elicit significant changes in foot net work (Fig. S3) or the free moment (Fig. S4)—thus, we cannot definitively rule out the possibility that work and/or free moment are related to foot temperature responses.

Blood flow mechanics have also been proposed to play a role in skin thermoregulation (Chatchawan et al., 2018; Walloe, 2016). The foot is primarily innervated by the posterior tibial artery which transverses the medial foot arch area (Sun, Jao & Cheng, 2005). It has been suggested that the foot temperature variation may be due to the overall effect of blood flow distribution within the foot where more distal structures (i.e., toes) are colder compared to the proximal structures of the foot (i.e., arch), in line with the innervation of the artery (Sun, Jao & Cheng, 2005). Thus arterial blood flow thermoregulation may have a larger influence within the arch area of the foot. However, it is also possible that arterial blood flow may have overall increased throughout the foot during turning, especially for the smaller radius condition that required increased medial-lateral shear forces to reorient the body’s direction of travel. It is also possible that venous blood flow may affect foot temperature (Hirata, Nagasaka & Noda, 1989). Changes in arterial or venous blood flow across the various radii conditions were not quantified in our study and future studies may need to explore the effect of blood flow on foot ΔT during walking.

While not included in our original hypotheses, gender differences in foot ΔT were observed in this study. Females had a lower baseline foot temperature compared to males, in line with other studies reporting the same trend in skin temperatures (Kim et al., 1998; Yasuoka et al., 2015; Jimenez-Perez et al., 2020). Temperature differences between genders may have occurred due to the contact area difference between genders, where males tend to have broader feet (Wunderlich & Cavanagh, 2001). This broad foot characteristic of males may have potentially allowed for lower shear stress during walking, or could have increased the contact area with a colder lab floor. However, we note that the findings regarding gender differences have limitations in that between-participants comparisons can be confounded by other external factors (e.g., nutrition, exercise prior to data collections, consumption of alcohol, etc.) (Moreira et al., 2017), which were not directly controlled for in this study. Future studies may need to control for these confounding factors to explore the relationship between gender and shear stress during walking.

Conclusion

In support of our hypotheses, we found a significant relationship between ΔT and resultant shear during continuous turning and toe-walking. We found that forefoot ΔT was more sensitive to resultant shear compared to the entire foot ΔT. We also found that there was a greater ΔT (specifically, less negative ΔT) within the external limb compared to the internal limb alongside greater shear. These results suggest that shear is an appropriate predictor of foot temperature responses. However, future studies should consider additional variables (e.g., blood flow, lab surface temperature, net work) that could influence the foot’s temperature response. Furthermore, the results of this study may motivate studying turning gait to quantify the relationship between shear and foot temperature in individuals who are susceptible to abnormal thermal regulations, such as in patients with diabetes.

Supplemental Information

Supplemental Information 1 Data

Click here for additional data file.

Supplemental Information 2 Thermal Imaging Checklist

A checklist was developed to determine external influences on our region of interest, developed by Moreira et al.,. The items in the checklist were assessed within our experiment procedures.

Click here for additional data file.

Supplemental Information 3 Temperature measurements were made utilizing a thermal imaging camera during pre- and post-walking conditions for all subjects (n = 18)

A 2 way ANOVA was utilized to determine differences in pre-temperature (baseline) between limbs and radii conditions. Baseline values increased with greater radius within the entire foot (P < 0.01) (Fig. S2 A) but not within the forefoot (P = 0.06) (Fig. S2 C). No significant differences were found between external and internal limbs in baseline temperatures within the entire foot (P = 0.90) nor forefoot (P = 0.46). No differences in post-temperature were determined between radius within the entire foot (P = 0.61) (Fig. S2C) or within the forefoot (P = 0.96) (Fig. S2D). No differences in post-temperature were determined between external and internal limbs within the entire foot (P = 0.06) nor forefoot (P = 0.10). (Values are means ± S.D.).

Click here for additional data file.

Supplemental Information 4 Average free moment (time-normalized to stance phase) (A) was extrapolated to the 5 min walking trial and utilized to gather free moment angular impulse (B) for all subjects (n = 18)

Free moment was normalized by body weight x height. Fig. S4 A demonstrates negative values when the limb has a clockwise moment (i.e., towards the turning direction) and positive values when the limb has a counter-clockwise moment (i.e., away from the direction of the turn). A 2 way ANOVA was utilized to determine differences between conditions and limbs where each were used as factors. Free moment angular impulse was not significantly different between limbs (p = 0.29) nor between radii conditions (p = 0.19). (Values are means ± S.D.).

Click here for additional data file.

Supplemental Information 5 Power distal to the foot (time-normalized to stance phase) (A) was integrated and extrapolated to the entirety of the 5 min walking trial to estimate total positive and negative work (B) as well as net work (C) for all subjects (n = 18)

Power was normalized by body mass. A 2 way ANOVA was utilized to determine differences in foot net work (Fig. S2C) between conditions and limbs. Net work was not significantly different between limbs (p = 0.92) nor between radii conditions (p = 0.97). (Values are means ± S.D.).

Click here for additional data file.

The authors would like to thank Jose Anguiano, Jenny Maun, and Nikos Papachatzis, members of the Department of Biomechanics, for their assistance in collecting the data.

Additional Information and Declarations

Competing Interests

Author Contributions

Human Ethics

Data Availability

The authors declare there are no competing interests.

Angel E. Gonzalez conceived and designed the experiments, performed the experiments, analyzed the data, prepared figures and/or tables, authored or reviewed drafts of the paper, and approved the final draft.

Ana Pineda Gutierrez performed the experiments, analyzed the data, prepared figures and/or tables, authored or reviewed drafts of the paper, and approved the final draft.

Andrew M. Kern conceived and designed the experiments, authored or reviewed drafts of the paper, and approved the final draft.

Kota Z. Takahashi conceived and designed the experiments, analyzed the data, authored or reviewed drafts of the paper, and approved the final draft.

The following information was supplied relating to ethical approvals (i.e., approving body and any reference numbers):

The study was conducted at the University of Nebraska at Omaha under the approval of the Internal Review Board of the University of Nebraska Medical Center (IRB Protocol #: 146-19-EP).

The following information was supplied regarding data availability:

Raw force plate data are available in the Supplemental Files.

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
