# Peer review of "Association between foot thermal responses and shear forces during turning gait in young adults"

_PeerJ, doi:10.7717/peerj.10515_

## Round 0.1 · original submission · Major Revisions

· Academic Editor

Major Revisions

Your letter has made me think that you could improve all the problems detected by our reviewers. What is also obvious to me that your work requires important changes, please make them in depth.

Reviewer 1 ·

Basic reporting

The aim of this work was to examine the shear-temperature relationship using turning gait with varying radii to manipulate magnitudes of shear onto the foot. The study is a very interesting and clearly written. The experiments are appropriated to analyze the research question. However, some aspects need to be addressed before to be considered to publication.

Experimental design

The research question was well defined and relevant. The investigation was adequate to assess this research question. More information is necessary for the methods;

Methodology, participants: Did the participants receive instructions related to nutrition, exercise the previous days, creams, alcohol, tobacco, etc., aiming to reduce the effect of confounding factors and ensure correct skin temperature measurements?

Methodology, line 153: Please, indicate the model of the infrared camera.

Methodology: More information about the infrared thermography protocol is necessary. I suggest to the authors to review this checklist (https://doi.org/10.1016/j.jtherbio.2017.07.006) to check which aspects are necessary to be provided in the manuscript.

Validity of the findings

No comment

Additional comments

In my opinion, the introduction was well written and perfectly explain the rationale of the study. However, I think that it is also important to mention the sweat and vasoconstriction/vasodilation mechanisms of the foot to better understand the foot skin temperature responses during walking.

Introduction, lines 40-42, applicable also for the second paragraph: I think that the authors need to specify better the sentence, describing distances, intensities, or characteristics of walking. This comment is because I have observed reductions in foot skin temperature after walking. A previous study stated this idea after barefoot walking 100 m at a comfortable speed (https://doi.org/10.1080/21681163.2018.1542349). Maybe authors also could show this contrast on the results between studies, which could support also the necessity of their study.

Methodology: It was used tympanic thermometer; however I am not sure if this data was used in the analysis. I suggest removing if this data was not used in the statistical analysis. Moreover, tympanic thermometer has important limitations in the measurement of core skin temperature (http://dx.doi.org/10.1016/j.jtherbio.2014.10.006).

Methodology, lines 154-156: It is interesting the methodology used to ensure consistent baseline temperatures. Do you have references of previous studies that used this methodology? I acknowledgment to the authors to have found a possible solution to the problem of performing several conditions in the same session on the effect of the order on baseline skin temperature. However, I have concerns about if the difficulty to do always in the same way this methodology could be a factor that explains the unexpected results of lower baseline skin temperature at the lower radius, which I think needs to be discussed.

Methodology, figure 2: I suggest to the authors to use the rainbow palette for the figure. This palette is more appropriate for human skin temperature. In addition, I suggest to the authors to modify the figure to represent the ROIs analyzed.

Methodology, line 197; Please, modify subjects by participants.

Methodology, secondary analyses, lines 224-246: It is unexpected that after statistical analysis appears more information regards the definition of more variables. Readers only could understand why authors include this information after reading all the results section. To clarify, I think that it could be better to include explanations of the variables before the statistical analysis section, and the extra analysis within the statistical analysis section without any subtitle.

Results, Figure 3 and 4: I would suggest to authors to reduce the extension of the figure captions and to write in a more systematic way. It is not necessary all these explanations that appear in other sections of the manuscript (e.g. statistical method used) or describing the results (e.g. Figure 2.B demonstrates…..). However, I would suggest the use of symbols in the figures to describes significant differences.

Discussion: Please, check the explanations about the decrease in skin temperature during walking mentioned in this reference (https://doi.org/10.1080/21681163.2018.1542349)

Discussion: In relation to the effect of gender on skin temperature, I would suggest authors check this recent publication (https://www.sciencedirect.com/science/article/pii/S0306456520300681)

Discussion: Please, discuss the practical applications of your results.

Reviewer 2 ·

Basic reporting

This article met the 5 “Basic Reporting” elements identified by PeerJ. Structure- and format-wise, the manuscript was laid out in a way that was easy to navigate. It was very clear the authors had spent time thinking about how to present and organize the data, and that the paper was very purposefully assembled.

Experimental design

I have mixed feelings with regards to the 5 “Experimental Design” elements identified by PeerJ. Divorced from the data, it all looks good. The authors explored and reported on many variables (both primary and secondary to the research goal), and I appreciated the thoroughness with which data was laid out. I understand/agree/empathize with the authors’ opinion (implied in lines 84-93) that, methodologically, standard force plates are tricky to use when trying to study lone shear forces. With all that said, when I consider the “Experimental Design” elements in light of the actual data, much of the data in unexplainable or unexpected, which suggests that something methods-wise wasn’t accounted for or some confounding factors interfered (explained in the following section).

Validity of the findings

My larger questions/concerns with the manuscript are with the data and its interpretation. I’m not a biomechanics expert and am focusing my review on the foot temperature components. What follows is a “thought map” to how I perceived the paper. Bullet-points are used here for communication purposes, but the four concepts are of course intertwined with each other, and some mental wrestling was needed to make this condensed list.
1. Foot temperature decreased from pre- to post-exercise (lines 288-299 and Figure 4), or rather, it decreased across males and stayed essentially the same across females (lines 349-354). Based on the majority of previous foot temperature studies, from multiple labs, one would expect foot temperature to increase during walking, regardless of sex, as the text discloses in lines 375-377.
2. The fact that baseline temperatures varied (lines 275-279, Figure S1) is also unexpected. The text says these results cannot be explained (lines 388-402). Since there are differences in baseline temperatures, but no differences in post-exercise temperatures (lines 284-286), this raises questions about the influence of the baseline temperature variation on the post-exercise temperature results.
3. Experimental procedures included artificially manipulating the subjects’ foot temperature, both through cooling (lines 154-156) and warming (lines 234-246). I understood that cooling happened once for each trial, but I couldn’t understand if the warming was done once or multiple times, or when it was performed relative to the other procedures. The text indicates the cooling procedure was to homogenize baseline temperatures (line 154), but that seems at odds with Thought #2 above. Regardless, this is a lot of artificial manipulation for an experiment that relies on “natural” foot temperature measurements, and may partially account for the results from Thoughts #1 and #2 above.
4. The floor was notably colder than subjects’ feet (lines 382-383 compared with lines 349-351 and 352-354). Although the paper initially states that statistics indicated no effect of floor temperature on changes in foot temperature (lines 384-386), the paper then hedges on that in lines 386-388 and again in 440-442. With regards to line 430, the paper states that males’ broader feet may have resulted in their experiencing lower shear forces; given the floor temperature, males’ broader feet also means increased surface area contact, and in turn greater ability to dissipate heat through the cooler floor, but this concept is not included in the text. The most parsimonious explanation appears to be that the cooler floor DID have an effect. I understand the chosen statistics suggest otherwise, but from a bird’s-eye-view of the data that doesn’t make sense.

Other minor concerns included: the sample size (line 125 says 24 subjects would supply 80% power, and this study included data from 18), especially in light of the sex effect; the use of tympanic temperatures as a measurement of core temperature (as these can be misleading); and the suggestion that the method can be used to identify people with conditions like diabetes or “neuropathic” conditions (as suggested in line 30).

Additional comments

Given all of these “moving parts”, I’m uncomfortable with the data, the conclusions, and possible generalizations (especially across sexes or to people with medical conditions). The wording of the paper may have played a role. The abstract only talks about increases in foot temperature (lines 14, 22, and 24), and doesn’t mention that post-exercise temperatures were lower than pre-exercise temperatures. Similarly, the portion of the Introduction related to foot temperature (lines 44-61 and line 71) also only talk about increases in foot temperature. When the reader gets to Figure 4, the results are quite different from what one would intuitively expect and what the paper has prepared the reader for. Phrases like “delta-T increased” (lines 288 and 292) are problematical. The abstract says a positive association between shear forces and temperature increase was found (lines 23-24), and the results indicate that smaller radii were associated with greater shear forces (254-255)…so, it’s less that “delta-T increased”, and more that smaller radii were associated with a smaller change in delta-T from pre- to post-exercise (radius 2 m has larger bars than radius 1 m because the temperature decrease was more blunted in the radius 1 condition than the radius 2 condition due to shear force thermal effects).

·

Basic reporting

No comment

Experimental design

No comment

Validity of the findings

No comment

Additional comments

To the Author:

General
The writing is crisp, clear and informed by a solid understanding of the topic and previous literature. It is a joy to read.

Title
The general guideline here is Population, Intervention, Outcome. If you can squeeze in information about the participants it would be excellent. Maybe include the phrase “young adults”?

Abstract
Succinct and comprehensive. I appreciate the inclusion of some actual data here.

Introduction
The argument developed is cogent and well structured. The logical flow of the literature support for the experiment is clear and convincing. This is so well written I have nothing to add, and this is an extremely rare event.

Methods
The methods, though complex and highly detailed, are well thought out and seem technically sound. The writing is again clear and complete, and each decision seems well rationalized. The experimental design would seem to induce dizziness, but other work on a 1m radius circle did not result in a deleterious or dangerous balance issue for participants. I believe the methods are sound, with one exception I do not fully understand…

Lines 234-246. The description of the shear stress calculation is somewhat confusing to me. I would like to know what the technical difficulties were, and if the MATLAB code has any reliability or validity data? How accurate was this method to determine sheer stress? Was this just a method to estimate area? It’s not clear.

Results
The results are clearly presented and very well polished. The progression through the variables of interest is appropriate and well ordered. It’s easy to follow. The integration of the graphical elements into the text is performed flawlessly. I think the addition of the df and F are generally unnecessary, but in this case I think it is best to include them. It makes the statistical results more comprehensive.

Graphs
The graphical elements are expertly crafted and illuminating. I appreciate the clean layout, and the appropriate ink density for the most important aspects.

Tables
--
Discussion
I think this treats the data quite well, and all reasonable confounding factors that may be attributed to the heating of foot tissue are discussed.

I think the venous return should be considered too, and not just arterial supply. It is more likely that an inability to remove excessive heat from the plantar surface of the foot contributes to foot temperature more than the arterial blood is warming the plantar tissue. After all, the sheer stress is supposed to be heating the foot from the mechanical friction. I think a little speculation here about venous return is warranted.

I was also hoping for some comment on the clinical utility or possible pathothermodynamics of the forefoot that might put a diabetic at greater risk of ulceration with elevated plantar tissue. Just how dangerous is the temperature increase you observed?

References
This seems comprehensive and includes the most salient literature.

---

## Round 0.2 · accepted · Accept

· Academic Editor

Accept

I am deeply sorry for my delay in be back to you. I am pleased with the effort you made in following all the reviewer's suggestions. I have no more comments. Thank you.

Reviewer 1 ·

Basic reporting

The only minor comment I should make is that figure 2 indicates "represent the areas to gather the entire foot (white)", and I think it is not white, but green.

Experimental design

The authors provided more details to the experimental design. The only doubt for authors and possibly for future readers is that basal temperatures were not the same, but authors have provided this as a limitation of the results of the study.

Validity of the findings

The presentation of the results was improved.

Additional comments

The authors have addressed all my comments satisfactory. Congratulations to the authors.